# Limitations and Modifications of Skin Sensitization NAMs for Testing Inorganic Nanomaterials

**DOI:** 10.3390/toxics12080616

**Published:** 2024-08-21

**Authors:** Britta Wareing, Ayse Aktalay Hippchen, Susanne N. Kolle, Barbara Birk, Dorothee Funk-Weyer, Robert Landsiedel

**Affiliations:** 1BASF SE, Experimental Toxicology and Ecology, 67057 Ludwigshafen, Germany; britta.wareing@basf.com (B.W.); ayse.aktalay-hippchen@basf.com (A.A.H.); susanne.kolle@basf.com (S.N.K.); dorothee.funk-weyer@basf.com (D.F.-W.); 2BASF SE, Agriculture Solutions, 67117 Limburgerhof, Germany; barbara.birk@basf.com; 3Pharmacy, Pharmacology and Toxicology, Free University of Berlin, 14195 Berlin, Germany

**Keywords:** inorganic nanomaterials, skin sensitization, OECD test guidelines, new approach methodologies, applicability, h-CLAT, LuSens, DPRA

## Abstract

Since 2020, the REACh regulation requires toxicological data on nanoforms of materials, including the assessment of their skin-sensitizing properties. Small molecules’ skin sensitization potential can be assessed by new approach methodologies (NAMs) addressing three key events (KE: protein interaction, activation of dendritic cells, and activation of keratinocytes) combined in a defined approach (DA) described in the OECD guideline 497. In the present study, the applicability of three NAMs (DPRA, LuSens, and h-CLAT) to nine materials (eight inorganic nanomaterials (NM) consisting of CeO_2_, BaSO_4_, TiO_2_ or SiO_2_, and quartz) was evaluated. The NAMs were technically applicable to NM using a specific sample preparation (NANOGENOTOX dispersion protocol) and method modifications to reduce interaction of NM with the photometric and flowcytometric read-outs. The results of the three assays were combined according to the defined approach described in the OECD guideline No. 497; two of the inorganic NM were identified as skin sensitizers. However, data from animal studies (for ZnO, also human data) indicate no skin sensitization potential. The remaining seven test substances were assessed as “inconclusive” because all inorganic NM were outside the domain of the DPRA, and the achievable test concentrations were not sufficiently high according to the current test guidelines of all three NAMs. The use of these NAMs for (inorganic) NM and the relevance of the results in general are challenged in three ways: (i) NAMs need modification to be applicable to insoluble, inorganic matter; (ii) current test guidelines lack adequate concentration metrics and top concentrations achievable for NM; and (iii) NM may not cause skin sensitization by the same molecular and cellular key events as small organic molecules do; in fact, T-cell-mediated hypersensitivity may not be the most relevant reaction of the immune system to NM. We conclude that the NAMs adopted by OECD test guidelines are currently not a good fit for testing inorganic NM.

## 1. Introduction

The applications of nanomaterials (NM) range through various fields, including coatings, cosmetics, medicine, and electronics [1]. As the application fields grew, extensive investigation of NM with respect to their potential effects on human health and the environment became necessary [2,3,4]. During the last decades, numerous publicly funded projects have generated methods and know–how on the characterization, toxicological testing, and grouping of NMs [5]. Since 2020, NM falls under the REACH regulations; thus, information on NM skin sensitization potential is required, as listed in Annex VII. Non-animal methods (or new approach methodologies, NAMs) are a priority for data generation under REACh, for small organic molecules and NM alike. Thereupon, an in vivo test shall only be performed if NAMs are not applicable to the test substance or deliver inconclusive results. 

In 2021, defined approaches (DA) to assess skin sensitization of test substances using data generated by NAMs were adopted as OECD guideline (GL) 497 [6]. The NAMs used as information sources in the DA address three key events of the adverse outcome pathway (AOP) for skin sensitization: (i) molecular interaction with skin proteins; (ii) activation of keratinocytes; and (iii) activation of dendritic cells. The first key event can be addressed by an in chemico assay, the direct peptide reactivity assay (DPRA). The DPRA is based on quantification of the depletion of artificial model peptides caused by the reaction of test substances with these peptides. The method was initially developed for testing small organic molecules with electrophilic functions, which can react with nucleophilic thiol and/or amino groups of cysteine and lysine of skin proteins (and also of the DPRA model peptides) [7]. Most inorganic NM are larger than most small organic molecules and do not hold distinct electrophilic functions, which would covalently bind to cysteine and/or lysine. Hence, the application of DPRA for testing inorganic NM is mechanistically questionable. The second key event, inflammatory response and changes in gene expression of keratinocytes, can be addressed, e.g., by the KeratinoSens^TM^ and LuSens assays with immortalized human keratinocyte cell lines. Both cell lines were genetically modified by inserting a luciferase reporter gene for monitoring the activation of the Keap1–Nrf2–ARE pathway. Luciferase signal increases due to Nrf2-mediated activation of antioxidant response element (ARE), initiated by oxidative or electrophile stress detected by the Keap-1 protein [8,9]. The third key event, the activation of dendritic cells, can be addressed by the so-called human cell line activation test (h-CLAT). The h-CLAT assay is performed with the human monocytic leukemia cell line THP-1 exposed to the test substance. The activation of these cells is measured via the changes in the expression of two surface markers, CD54 and CD86, stained with fluorescent antibodies and quantified in a flow cytometer [10]. 

According to the first DA of the OECD GL No. 497, the “2 out of 3” (2o3) DA, any two of the three tests addressing these key events of the AOP determine the overall results. The second DA, the ITSv2 DA, uses information from the DPRA and h-CLAT combined with protein binding alerts generated using the OECD QSAR Toolbox. The overall result of the DA is discerning potential skin sensitizers and non-sensitizers (hazard assessment) in accordance with the UN GHS sensitizer categories. The DA GL based on AOP for skin sensitization was initially adopted to assess the skin sensitization potential of small organic molecules [4,11,12]. 

Thus far, only a few studies on NM with skin sensitization NAMs have been conducted. Likewise, only a few animal studies on NM’s skin sensitization are available [13,14,15,16,17,18,19,20]. Physical and chemical properties of inorganic NM differ from those of small organic molecules. Most NMs are larger and often not monodisperse. There is no defined molecular weight of NM and a distribution of different particle sizes [1]. Larger NM penetrate only a few layers into the stratum corneum of intact skin but do not become bioavailable; thus, no interaction with living cells is expected [21,22]. Most inorganic NMs would interact with proteins via mechanisms other than covalent binding, e.g., non-covalent metal coordination bonds, or redox reactions. Moreover, inorganic NM can release metal ions and generate reactive oxygen species (ROS). NM forms protein coronas, which could alter the structures of the proteins. These could be molecular initiating events related to altered immune functions [3,11,23,24,25,26,27,28], but they may not be well captured by the current NAMs. In addition to these mechanistic stipulations, there are also technical hurdles: NMs are not directly applicable to these NAMs as they are not soluble (as long as they are NM), which requires a test item preparation different from the dissolution of small organic molecules and modifications to avoid interferences with optical read-outs of the NAMs.

To assess the applicability of the NAM-based DA for skin sensitization, we have tested nine widely used inorganic NM with three skin sensitization NAMs, which were adopted as OECD TGs (No. 442C, D, and E).

## 2. Materials and Methods

NM and quartz (SiO_2_ DQ12) were tested in three NAMs: DPRA, LuSens, and h-CLAT (OECD TGs No. 442C [29], D [30], and E [31]) with modifications in test substance preparation (described below) to adapt these three assays for their applicability to NM. The results of these three assays were evaluated based on the criteria of DA OECD GL No. 497 [6], i.e., including borderline ranges [32].

### 2.1. Nanomaterials

In the present study, eight inorganic NM and one micron-sized material (SiO_2_ DQ12) were assessed (Table 1). They consist of different metals or metalloids (Ba, Ce, Si, Ti, and Zn) as oxides or sulfates and represent different sizes and crystallinities (SiO_2_ Aerosil, Levasil, and DQ12). 

### 2.2. Test Substance Preparation 

The preparation of the test substances was carried out based on the procedures described by the NANOGENOTOX standard operation procedure (SOP). This dispersion protocol was developed in the EU-project NanoREG and is one of the most widely used dispersion protocols. It was developed for the preparation of general batch dispersions for in vitro and in vivo toxicity testing to produce a highly dispersed state of a wide range of nanomaterials [35,36]. A fixed (gravimetric) test substance concentration of 2.56 mg/mL was prepared for every NM. Moreover, 25.6 mg of the respective test substance were weighed into a glass vessel and pre-wetted with 50 µL of ethanol. Thereafter, 9.95 mL of the respective vehicle (bovine serum albumin (0.05 *w*/*v*%) in deionized water for DPRA and cell culture medium for LuSens (with additional 4% DMSO) or h-CLAT assay) was added. The stock preparation was sonicated using a Branson Sonifier SFX 550 with 20% amplitude (400 W) for 16 min. Stock preparations were thoroughly shaken using a vortex prior to following dilution steps, and the homogeneity of the dispersion was assessed by visual inspection. All test substance stocks were prepared freshly for each experiment. The preparations were generally applied within two hours after weighing the test substances; the test substance preparations were homogenous and stably dispersed by visual inspection unless otherwise noted for individual assays.

### 2.3. Test Concentrations

The stock dispersions of the test substance were prepared at 2.56 mg/mL as described above. The test guidelines prescribe either test substance concentrations with a defined molar concentration or a so-called “gravimetric approach“ using mass per volume concentrations. Also, the constituents of NMs have a molar mass (BaSO_4_ 233, CeO_2_ 172, SiO_2_ 60, TiO_2_ 80, and ZnO 81 g/mol), but at a given molar concentration, most of the test substance is not dissolved but only dispersed, i.e., packed inside the particles. Accordingly, the “gravimetric approach” could be applied by using an assumed average molecular weight of 200 g/mol for test substances without defined molar mass—but still assumed to be dissolved. Since this assumption does not hold, instead the relevant concentration metric of NM and other solid particles would be different; i.e., particle number per volume or particle surface area per volume have been proposed [37,38,39]. In addition, the nominal concentration of a NM in cell culture media (even for stable dispersions and using any concentration metric) may not be the effective concentration [40,41,42]. For other reasons, this is also true for dissolved molecules [43,44]. While the respective test guidelines define nominal maximum concentrations to be tested, the maximum concentration may be limited by cytotoxicity as described in the guidelines for test systems using living cells (LuSens and h-CLAT) and elsewhere [45,46,47]. Preparing dispersions of NM in cell culture media can greatly influence the results of in vitro tests [48]. Therefore, the dispersion protocols were standardized. The maximum concentration of a NM in a stable cell culture media dispersion is limited. We implemented one of the established protocols [35], which is using a maximum concentration of 2.56 mg/L. As a consequence, the maximum concentration of NM in the respective test system resulted from the dilution of this stock dispersion according to the respective protocols.

DPRA: Dilutions of test substances were prepared from the stock preparation (2.56 mg/mL) according to the standard pipetting procedure given in the DB-ALM protocol [49] (nominal, based on the constituents of the respective NM: 2.7 to 10.6 mM instead of stock solution concentration of 100 mM or 20 mg/mL for soluble organic test substances as required in OECD TG No. 442C) dilutions for C-peptide and K-peptide incubations, respectively, with 0.50 mM of the corresponding peptide.

LuSens: Dilutions of test substances were prepared from the stock preparation (2.56 mg/mL) with 640 µg/mL as the maximum concentration (while the TG requires a concentration of 2000 µM or 2000 µg/mL). Test substance concentrations were adjusted if cytotoxicity was observed (ZnO, CeO_2_, both of the TiO_2_ NM, SiO_2_ Levasil 200, SiO_2_ Aerosil 200, and SiO_2_ DQ12). Tested concentrations are given in the Appendix A. Consequently, only BaSO_4_ and SiO_2_ Aerosil R972 were tested at concentrations that were not cytotoxic and below 2000 µg/mL. Higher test concentrations were, however, not technically achievable with the concentration of 2.56 mg/mL of the stock dispersion, which was prepared according to the standardized protocol.

h-CLAT: The 1st experiment was always conducted with a maximum concentration of 1280 µg/mL, corresponding to the highest applicable dilution from the 2.56 mg/mL test substance stock preparation (according to the test guideline, the top final concentration is 1000 µg/mL). If this is non-cytotoxic, the maximum concentration should be re-determined. In any case, the final concentration in the plate should not exceed 5000 µg/mL for test chemicals dissolved or stably dispersed in saline or medium.

At the 1st experiment, ZnO, TiO_2_ NIST^®^ SRM^®^, TiO_2_ Aeroxide P25, CeO_2_ (all <50% viability), and BaSO_4_ (≤80% viability) were cytotoxic at the maximum concentration of 1280 µg/mL. The test concentrations were lowered accordingly in the following experiments. All SiO_2_ test substances were not cytotoxic at the maximum concentration of 1280 µg/mL (viability ≥90%). The concentrations were, however, not increased in the following experiments as higher test concentrations were not achievable with the stock dispersion, which was prepared according to the standardized protocol. Cytotoxicity data and tested concentrations are given in the Appendix A. 

### 2.4. Direct Peptide Reactivity Assay (DPRA)

The DPRA experiments were performed following an adapted protocol based on OECD TG No. 442C. Following the stock preparations according to the above-described NANOGENOTOX protocol, the test substance dilutions were prepared (as triplicates) following the standard pipetting procedure (given in the DB-ALM protocol No. 154): (1) stock preparation was diluted 20-fold and incubated with 0.50 mM of the cysteine-containing peptide (C-peptide) (Ac-RFAACAA-COOH, 752 g/mol), and (2) stock preparation was diluted 1:4 and incubated with 0.50 mM of the lysine-containing peptide (K-peptide) (Ac-RFAAKAA-COOH, 776 g/mol) for 24 h at approx. 25 °C. The remaining non-depleted peptide concentration was determined by high-performance liquid chromatography (HPLC) with gradient elution and UV detection at 220 nm (corresponding to the absorption band of phenylalanine). In parallel, triplicates of the concurrent vehicle control and a positive control (PC) (50 mM ethylene glycol dimethacrylate (EGDMA)) were incubated with the peptides. Additionally, a co-elution control was carried out to detect possible interference of the test substance due to co-elution with the peptides via analysis of the ratio of peak areas at 220 nm and 258 nm. The mean depletion was evaluated based on TG No. 442C and GL No. 497 with criteria listed below (Table 2), defining negative, borderline, or positive reactivity.

### 2.5. LuSens Assay

The LuSens assay was performed following an adapted protocol based on OECD No. TG No. 442D. The assay consisted of two parts: first, a cytotoxicity pre-experiment for determining the CV75 value (estimated concentration that affords 75% cell viability by linear regression) and a basis to determine the concentration range for the main experiment. If any interference between the test substance and MTT read-out was observed in the absorption spectrum at OD550nm, an alternative viability assay (CellTiter-Glo^®^, Promega GmbH, Walldorf, Germany) was performed to quantify ATP. Second, the main experiment consisted of measurement of the luciferase induction after 48 h exposure time (as triplicates) to the test substance with at least 8 concentrations for determining the keratinocyte activation. The maximum concentration tested in the main experiments was 640 µg/mL. For this, the cells were lyzed, and luciferase induction was evaluated by measuring luminescence signal after substrate addition (One Glo^®^, Promega). Cell viability was determined for each tested concentration with either an MTT or ATP assay. In parallel, triplicates of the concurrent vehicle control, a positive control (PC) (18 µg/mL ethylene glycol dimethacrylate (EGDMA)) and a negative control (450 µg/mL DL-lactic acid (LA)), were incubated with the cells. At least two independent experiments were performed for evaluation of the luciferase induction. Based on the OECD TG No. 442D and borderline ranges [32,50], mean fold induction in at least two independent experiments was evaluated with criteria listed below (Table 3) to conclude a negative, borderline, or positive keratinocyte activation potential. 

### 2.6. h-CLAT Assay

The h-CLAT assay was performed following a modified protocol based on the OECD TG No. 442E with adaptations for NM implemented, as described above for the test substance preparation, and as follows. No pre-test for cytotoxicity assessment was performed. The 1st main experiment was always conducted with a maximum concentration of 1280 µg/mL, corresponding to the highest applicable concentration from the 2.56 mg/mL test substance stock preparation. For the main experiments, the human cell line THP-1 was treated with at least 8 testsubstance concentrations for 24 h. In parallel, the concurrent vehicle control, a positive control (PC) (4 µg/mL 1-chloro-2,4-dinitrobenzene (DNCB)), and a negative control (1000 µg/mL DL-LA) were incubated with the cells. To assess the matured cell population by flow cytometry density gradient centrifugation (on a Ficoll-Paque PLUS (GE Healthcare, 17-1440-02) with centrifugation at 1500 rpm for 15 min at room temperature, without break function in the deceleration phase) was performed to remove the test substance before applying to the flow cytometer to avoid interferences and damages to the instrument. Thereafter, cells were stained with PI along with FITC-labeled anti-human CD86 antibodies or FITC-labeled anti-human CD54 antibodies or the corresponding isotype control FITC-labeled anti-mouse IgG1. The relative mean fluorescence (RFI) of CD86 and CD54 was evaluated by measuring the CD86 and CD54 signals by flow cytometric analysis. In parallel, the cytotoxicity of the test substance was measured using the fluorescence intensity of PI. At least two independent experiments were performed for evaluation of the dendritic cell activation, and test-substance concentrations were adjusted if evaluation criteria for viability (>50% in at least four tested concentrations) of an experiment were not met. The mean RFI induction in two independent experiments was evaluated based on OECD TG No. 442E and OECD GL 497, with criteria listed below (Table 4) defining a negative, borderline, or positive dendritic cell activation. 

### 2.7. Defined Approach

The results obtained in three NAMs were evaluated according to the AOP-based “2 out of 3” (2o3) and ITSv2 DAs for skin sensitization hazard identification described in OECD GL No. 497. In 2o3, any two of the three tests determine the overall results, i.e., if any two assays result positive, the overall test results yield the prediction of a sensitizer, while any two negative test results would yield the prediction of a test substance to be a non-sensitizer. To include experimental uncertainties for the predictivity of single assays, borderline ranges were considered for the evaluation of skin sensitization potential. In ITSv2, scores are assigned for the results of DPRA and h-CLAT experiments and for the results from the OECD QSAR Toolbox to yield an overall result that predicts the skin sensitization potency of chemicals: UN GHS category 1A (strong sensitizer), category 1B (other sensitizer), or not classified (non-sensitizer). 

## 3. Results and Discussion

Nine particles, eight inorganic NM and DQ12, were tested in DPRA, LuSens, and h-CLAT assays. These are insoluble (as long as they are particles) and require modifications of the test protocols.

### 3.1. DPRA

The DPRA experiments were performed with a modified protocol based on OECD TG No. 442C, originally described by Gerberick et al. in 2004 [7]. The non-depleted peptide concentrations were determined by HPLC, and the depleted peptide concentrations evaluated (Table 5).

Mean peptide depletion was <2% for all nine test substances (at a stock concentration of 2.56 mg/mL). The tested stock concentration was below the required concentration (20 mg/mL in the gravimetric approach) and hence the ratio to the peptide according to TG442C [29]. The molar mass of the elemental composition of the selected NM was used to calculate the nominal concentrations, and the resulting test substance-to-peptide ratios were calculated based on the nominal concentrations.

Nevertheless, these “negative” or non-reactive results have to be considered “inconclusive” according to TG No. 442C [29], due to the following reasons: (1) Metals and inorganic substances should not be tested in DPRA, as they are “known to react with proteins via mechanisms other than covalent binding”. (2) Insoluble substances cannot deliver firm conclusions on the lack of reactivity if no reactivity is observed, since it is “not sure if sufficient exposure can be achieved” and the effective concentration is unknown, in particular if sedimentation occurs despite prior sonification [51]. Studies on few inorganic materials have however demonstrated that indeed these materials can also yield positive results in the DPRA (compiled in [52]) or a modified version of the assay using LC-MS/MS detection (but adducts were not analyzed) testing multiple concentrations [53]. Further, Roberts et al. (2007) describe transition metals as being able to covalently bind proteins via covalent coordination bonds [54], and in a recent study with inorganic NM, positive results for C-peptide reactivity (and no reactivity for K-peptide) were reported for TiO_2_ and CeO_2_ nanoparticles [13,15]. The C-peptide, however, is prone to dimerization by oxidation, which would lead to peptide depletion without binding of the test substance, leading to an overestimation of reactivity, and metal ions had been described to cause dimerization of the thiol-group-containing nucleophile in the ADRA [55]. Therefore, positive results with thiol nucleophiles should be evaluated with caution, and the formation of reaction products of the peptides should be quantified to the depletion of the peptide. An approach to differentiate between dimers and reaction products has been proposed for the ADRA to identify false positives due to thiol nucleophile dimerization [56]. While no reactivity was observed for any of the NM assessed in the present study, adduct analysis and kinetics may clarify the applicability domain limitation stated in OECD TG No. 442C.

With adaptations, the DPRA was technically applicable to insoluble NM. According to the method’s data interpretation procedure (DIP), it can only deliver conclusive results for non-dissolved test substances in the case of a peptide depletion above the defined threshold, whereas less peptide depletion is not assessed as being negative, but as inconclusive. To enable the investigation of inorganic test substances, which can react with proteins via mechanisms other than covalent binding, different analytical methods are required that can address these reactions and identify potential adducts (e.g., metal complex formation) or modified proteins in the corona of the NM. 

### 3.2. LuSens

The LuSens experiments were performed following a modified protocol based on the OECD TG 442D, originally described by Ramirez et al. in 2014 [30]. For cytotoxicity measurements, MTT tests with a range of dilutions were performed. For CeO_2_ NM-212, prior centrifugation was necessary, as large deviations were observed in MTT measurements. For SiO_2_ DQ12 and TiO_2_ NIST^®^ SRM^®^ ATP tests were conducted instead of MTT tests as an alternative cytotoxicity determination. After assessment of cytotoxicity, appropriate test substance concentrations were selected for incubations to assess luciferase induction. The resulting CV75 and EC 1.50 values were calculated, and keratinocyte activation was evaluated (Table 6). 

Four NM yielded negative results, whereas two NM and DQ12 were tested positive. Three replicate experiments with BaSO_4_ delivered inconsistent results: a negative and borderline result and an invalid experiment due to high cytotoxicity. Experiments with SiO_2_ Aerosil R972 delivered borderline results; however, these results were considered inconclusive due to visible inhomogeneity and the test guideline requiring a homogenous test substance preparation. Overall, the LuSens assay was technically applicable with adaptations to the TG protocol. 

Despite the conclusive results according to TG442D, these results should be interpreted with caution, as the predictivity of the LuSens assay is based on the reaction of the test substance (or products formed from or by the test substance) with thiols of Keap-1, which is then activating the Keap1–Nrf2–ARE pathway [24]. The reactivity of solid, inorganic materials with these thiols is not well investigated. Interestingly, none of the NM exhibited any significant reactivity in the DPRA with the C-peptide containing a thiol group, while three test substances (SiO_2_ Levasil 200 (40%), SiO_2_ DQ12, and ZnO Z-Cote HP1) delivered clearly positive results in the LuSens assay. Two other SiO_2_-based NM were tested negative and borderline; hence, no direct correlation of elemental composition could be driven, and the published data on SiO_2_-based NM are also not consistent [17,57]. Recently, studies with NM in the KeratinoSens assay (addressing the same key event as the LuSens assay) were conducted. In one study, positive results for gold and silver nanoparticles were reported, whereas NiO and TiO_2_ were tested negative [20]. In another study, CuO was tested positive, whereas CeO_2_, TiO_2_, ZnO, and other metal oxides (NM) were tested negative [16]. The negative results for the TiO_2_-based NM and CeO_2_ in our experiments agree with these studies’ outcomes. For ZnO, DQ12, and SiO2 Levasil, our experiments indicated an activation of the Keap1–Nrf2–ARE pathway. This pathway is contributing to skin sensitization in the context of other KEs of the AOP. Independent of this, activation of this pathway indicates oxidative stress, which can indeed be caused by inorganic NM, either by catalyzing the formation of ROS or by releasing metal ions. Previous studies investigated the surface reactivity and ROS production of NM [23,25] by different methods (electron spin resonance (ESR) spectroscopy, ferric reducing ability of serum (FRAS), and protein carbonylation). The relative activity of one NM compared to another varied with different methods, and some nanoforms of a NM showed results different from another nanoform [58]. In several assays, DQ12, ZnO, and some silica showed higher activities (based on the same mass concentration) than BaSO_4_, CeO_2_, and TiO_2_ NM. The LuSens assay may serve as another assay to assess the surface reactivity and ROS production of NM, which is one criterion to group NM [59,60] and define sets of similar nanoforms [58,61]. This approach has already been explored [23]. 

### 3.3. h-CLAT

The h-CLAT experiments were conducted following a modified protocol based on the OECD TG No. 442E, without cytotoxicity measurements prior to the main experiment. Hence, the first main experiment was always conducted with the highest achievable concentration (diluted from the stock preparation), and the concentration was adjusted after assessment of cytotoxicity. After 24-h exposure time, density gradient centrifugation was performed to separate NM from cells prior to flow cytometry to avoid interferences. To investigate the influence of this step, experiments with and without centrifugation were conducted with SiO_2_ Levasil 200 (40%). No apparent influence of centrifugation on cell viability was observed while particles were removed. Cells were stained with the corresponding fluorescently labeled antibodies. The relative mean fluorescence (RFI) of CD86 and CD54 were measured, which were then used to calculate EC150 and EC200 values for CD86 and CD54, respectively, and evaluated to assess the dendritic cell activation potential of NM (Table 7). 

Two NM and DQ12 yielded clearly positive results for one of the CD markers, whereas only one NM, BaSO_4_, was tested negative for the upregulation of both surface markers. Most of the NM yielded negative or borderline results for one of the surface markers. According to DIP of the TG, the tests should be performed up to a defined concentration (2 mg/mL) or up to the cytotoxicity limit. However, application at this concentration was not possible following the NANOGENETOX protocol. Hence, all NM should have been tested at a concentration limited by their cytotoxicity. Three NM, SiO_2_ Aerosil 200, SiO_2_ Aerosil R972, and SiO_2_ Levasil 200, yielded negative or borderline results for CD54; however, in these experiments, the cytotoxicity limit was not reached; thus, the results are considered inconclusive. A previous study with silica NM investigated with LuSens and h-CLAT assays found no activity in either assay and not cytotoxicity even at the highest concentrations [17]. In another assay adopted as part of OECD TG 442E, namely the GARDskin assay, 13 metal compounds were tested, and the others concluded a similar predictive capacity as compared to defined organic molecules [62].

In conclusion, the h-CLAT assay was technically applicable with the addition of the centrifugation step to the TG protocol. The significance of negative or borderline results obtained with this assay is, however, limited by the achievable maximum concentration with a given dispersion protocol. Moreover, the biological relevance of this assay towards the potential skin sensitization of NM is questionable.

### 3.4. Assessment of the Skin Sensitization Potential with DAs

The results obtained in three NAMs (according to OECD TG No. 442C, D, and E), addressing three KE in skin sensitization AOP, were evaluated according to the 2o3 and DA for skin sensitization described in OECD GL No. 497. First, all results obtained in the three assays were considered in the evaluation (Table 8), even though some are inconclusive according to the TG since the test substances are inorganic materials, are not completely soluble, and/or sufficient test concentrations of dispersed particles could not be achieved by standard protocols (please see footnotes for details). 

In the 2o3 DA, of the nine materials tested, two NM were evaluated inconclusive due to solubility/inhomogeneity issues (SiO_2_ Levasil 200 (40%) and SiO_2_ Aerosil R972), and two (SiO_2_ DQ12 and ZnO Z-Cote HP1) were identified as skin sensitizers. The remaining five NM possibly have no skin sensitization potential; however, the final evaluation has to be considered inconclusive because (i) negative results of the DPRA with inorganic and insoluble test substances are inconclusive, and (ii) LuSens and h-CLAT assays partly delivered inconclusive results due to borderline results and inconsistencies between single experiments. 

OECD GL497 also provides DAs comprising the DPRA, h-CLAT, and the OECD toolbox as information sources in ITSv2. Using the present dataset, evaluation according to ITSv2 DA was challenging due to negative/inconclusive results in the experimental information sources and the unavailability of protein binding alerts from the OECD toolbox. Two NM could be evaluated with ITSv2. They are assessed as being skin sensitizers by this DA. Both DAs consistently indicate skin sensitization potential for DQ12 and ZnO NM. Both test substances tested negative in animal studies, and ZnO was also found negative in humans [63,64]. However, both materials have substantial ROS-production capabilities, and the results obtained with NAMs may rather reflect this activity than specific skin sensitization potential [23,25]. According to in vivo skin sensitization studies (TG No. 442B: Skin Sensitization Local Lymph Node Assay, LLNA: BrdU-ELISA), TiO_2_, CeO_2_, SiO_2_, and ZnO NM do not induce skin sensitization in the mice [17,19,65]. LLNA data of small organic molecules are, however, not well correlated to skin sensitization in humans, and in vitro methods should be assessed towards human relevance [66]. Moreover, with the LLNA, test substances are applied to the intact skin, and the negative results could merely reflect the absence of sufficient skin penetration. Bypassing the skin barrier for a more conservative hazard assessment would require a Guinea Pig Maximization Test using an adjuvant to increase skin penetration [67]. This test has been performed for CeO_2_ NM-212, SiO_2_ Levasil 200, materials related to SiO_2_ R972, and TiO_2_ P25; all four NM were found negative (see Table 1 for references). 

A principal mechanism by which solid NM could cause skin sensitization is via the release of ions (or other soluble substances) with a skin sensitization potential. None of the ions constituting the NM tested in this study are classified as skin sensitizers. However, dissolution of skin sensitizing constituents from respective NM (e.g., nickel nanoparticles) could give relevant info towards a potential for skin sensitization. The solubility can give first-tier information on the expected concentration of released ions; the solubility values of the nanoparticles in this study are listed in Table 1. More refined methods to measure dissolution kinetics of NM are available [68].

## 4. Conclusions

NM could technically be applied to skin sensitization NAMs (DPRA, LuSens, and h-CLAT assays) with some adaptations. The dispersion of the NM is crucial. In this study, NM dispersions were prepared using a standardized protocol to obtain highly dispersed NM, which were applied to the test systems directly after preparation. Some of the TG are using molar concentrations that are not defined for NM. Gravimetric approaches in the TG are based on mixtures and formulations consisting of small molecules with the assumption of an average molecular weight of 200 g/mol, thus not directly applicable for larger structures such as solid particles. An adequate concentration metric for NM testing in skin sensitization NAMs needs to be defined.

None of the NM-depleted peptides in DPRA. Nevertheless, all results had to be considered inconclusive due to the insolubility of the NM and because inorganic materials are out of the applicability domain of the DPRA.

In cell-based assays, further adaptations were required due to interference with the read-out. Centrifugation steps were performed to separate NM from cells prior to optical read-out. Some NM interfered with the MTT assay, and ATP assays were used as an alternative for viability measurement; PrestoBlue assays could be another alternative [69]. Three test substances were positive in the LuSens assay. These test substances showed a higher ROS production in other assays, and the LuSens assay may well be suitable to detect NM-induced ROS formation in mammalian cells. Three test substances were positive in the h-CLAT assay for dendritic cell activation. Two of them, ZnO and DQ12, were also positive in the LuSens assay. Consequently, these two substances were assessed as overall positive according to the DA. They were, however, not showing skin sensitization potential in animal studies nor—for ZnO—in humans. For seven NM, no consistent overall assessment could be derived from the three NAM tests. All are inorganic and insoluble and hence outside the applicability domain of the DPRA. Four of them did not form stable dispersions or did not achieve the maximum concentrations defined in the TG of the LuSens or h-CLAT. Three NM gave borderline results in the h-CLAT or non-consistent results in the LuSens and h-CLAT. For all NM, the current NAM TG does not define adequate concentration metrics. Moreover, the relevance of the tested KE towards skin sensitization by inorganic, insoluble matter is unknown. Relevant mechanisms of NM skin sensitization should be identified, and respective NAMs (with adequate concentration metrics) should be developed and validated for reliability and human relevance (ideally using human reference data). The current NAMs, which were developed for small organic molecules, seem not to be a good fit for NM. The currently available data from humans, animal studies, and in vitro do not indicate a skin sensitization potential for inorganic NM without skin sensitizing constituents.

## Figures and Tables

**Table 1 toxics-12-00616-t001:** Test substances and their properties.

Test Substance	CAS-Number	Physicochemical Properties	Skin Sensitization Potential
CeO_2_ NM-212	1306-38-3	40 nm (SEM) very low water solubility (<0.001 wt.%) ^#^	Cerium oxide is reported non-sensitizing in the GPMT [33]
BaSO_4_ NM-220	7727-43-7	25 nm (SEM) ^#^ low water solubility (<0.05 wt.%) ^#^	bulk material was reported non-sensitizing in the LLNA ^1^
SiO_2_ Levasil 200 (40%)	7631-86-9	5–50 nm (REM/TEM) * soluble in water	reported non-sensitizing in the GPMT * and the LLNA ^1^
Quartz SiO_2_ DQ12	n.a.	500–750 nm * crystalline quartz practically insoluble *	reported non-sensitizing in the LLNA ^1^
SiO_2_ Aerosil R972	68611-44-9	16 nm ^+^ hydrophobized colloidal silica water solubility > 1 mg/L *	Aerosil R812 and R 8200 were reported non-skin sensitizing in the GPMT ^1^
SiO_2_ Aerosil 200	7631-86-9	9 nm (TEM/SEM) [25] hydrophilic fumed silica water solubility > 1 mg/L *	bulk material was reported non-sensitizing in the LLNA ^1^
TiO_2_ NIST^®^ SRM^®^	13463-67-7	19–37 nm ^+^ very low water solubility (0.001 g/L) *	reported non-sensitizing in the LLNA *
TiO_2_ P25 Aeroxide	13463-67-7	21 nm * hydrophilic titanium dioxide	reported non-sensitizing in the patch test and the GMPT *
ZnO Z-Cote HP1	1314-13-2	190 nm [34]	ZnO nanomaterials were reported as non-sensitizing in patch test ^1^

* Data provided in the safety sheet or certificate of analysis. ^#^ Value determined in the Department of Analytical and Material Science, BASF SE. ^+^ Value given in the manufacturer’s website. ^1^ Data published in the registration dossier released by ECHA.

**Table 2 toxics-12-00616-t002:** Evaluation criteria of DPRA, according to OECD TG No. 442C and GL No. 497, based on the C-peptide/test substance 1:10/K-peptide/test substance 1:50 prediction model (mean peptide depletion) and the C-peptide/test substance 1:10 prediction model (C-peptide-only depletion).

Mean Peptide Depletion [%]	C-Peptide Depletion [%] ^1^	Reactivity	Evaluation
>42.47	>98.24	high reactivity	positive
>22.62; ≤42.47	>23.09; ≤98.24	moderate reactivity	positive
>6.38; ≤22.62	>13.89; ≤23.09	low reactivity	positive
>4.94; ≤8.32	>10.55; ≤18.47	borderline ^2^	inconclusive
≤6.38	≤13.89	minimal or no reactivity	negative ^3^

^1^ If mean peptide depletion [%] could not be determined due to invalid K-peptide depletion (e.g., insolubility of the K-peptide samples or interference in the samples of the K-peptide), C-peptide depletion is considered for evaluation. ^2^ According to OECD GL No. 497, results in this range were considered borderline and evaluated as inconclusive due to their uncertainty. ^3^ For test substances that were not completely soluble by visual observation in the sample preparations containing the peptides immediately after preparation or after 24 h, the result may be under-predictive due to limited availability of the test substance. In this case, if the mean peptide depletion was below the borderline range, the reactivity was considered “inconclusive”. Likewise, is a mean peptide depletion was above the borderline range, a test substance was considered “positive”.

**Table 3 toxics-12-00616-t003:** Evaluation criteria of LuSens assay according to OECD TG No. 442D and GL 497.

Mean Fold Induction	Reactivity	Evaluation
>1.76	activates keratinocytes	positive ^1^
>1.28; ≤1.76	borderline ^2^	inconclusive
≤1.28	does not activate keratinocytes	negative ^3^

^1^ If the luciferase induction was above 1.76-fold, statistically significant in at least 2 consecutive tested concentrations with a cell viability greater than 70%, and at least 3 test concentrations were non-cytotoxic, the outcome of the LuSens assay was considered positive. ^2^ Applying borderline range criteria for the LuSens assay [32,49], such as described in OECD GL 497, results in this range were considered borderline and evaluated as inconclusive due to their uncertainty. ^3^ A test substance was considered negative when the criteria mentioned above were not met up to the maximum concentration of 2000 µg/mL or up to the cytotoxicity limit (at least one concentration displaying viability below 70%).

**Table 4 toxics-12-00616-t004:** Evaluation criteria of the h-CLAT assay according to OECD TG No. 442E and GL 497.

Mean RFI (CD86)	Mean RFI (CD54)	Reactivity	Evaluation
>184	>255	activates dendritic cells	positive ^1^
>121; ≤184	>156; ≤255	borderline ^2^	inconclusive
≤121	≤156	does not activate dendritic cells	negative ^3^

^1^ If the mean RFI value for CD86 was greater than 184 and/or the mean RFI value for CD54 is greater than 255 at any concentration in relation to vehicle control that does not reduce viability below 50% and reproduces in the same cell surface marker in at least an additional independent experiment, the h-CLAT assay was considered positive. ^2^ According to OECD GL 497, results in this range were considered borderline and evaluated as inconclusive due to their uncertainty. ^3^ A test substance was considered negative when the ‘positive’ criteria mentioned above were not met up to the maximum concentration of 5000 µg/mL for the vehicle culture medium or 1000 µg/mL for 0.2% DMSO in culture medium or up to the cytotoxicity limit (with viability less than 90% at the highest concentration tested). If these criteria were not met, no conclusion on dendritic could be derived under these test conditions.

**Table 5 toxics-12-00616-t005:** Results and evaluation of the DPRA with test substances. Peptide depletion values for K-peptide and C-peptide, as well as the mean of both peptides, are given.

Test Substance	K-Peptide Depletion [%]	C-Peptide Depletion [%]	Mean Depletion [%] ^1^	Result ^2^
CeO_2_ NM-212	0.60 ± 0.46	3.28 ± 0.94	1.94	negative ^3^
BaSO_4_ NM-220	0.00 ± 0.24	1.96 ± 0.82	0.98	negative ^3^
SiO_2_ Levasil 200 (40%)	0.25 ± 0.80	−0.36 ± 0.28	0.12	negative ^3^
SiO_2_ DQ12	−0.41 ± 0.97	−0.20 ± 0.64	0.00	negative ^3^
SiO_2_ Aerosil R972	−0.30 ± 0.63±	−2.00 ± 1.01	0.00	negative ^3^
SiO_2_ Aerosil 200	−0.55 ± 0.33	−0.88 ± 1.44	0.00	negative ^3^
TiO_2_ NIST^®^ SRM^®^	−2.62 ± 0.84	−1.67 ± 2.67	0.00	negative ^3^
TiO_2_ P25 Aeroxide	−2.91 ± 0.39	1.16 ± 2.53	0.58	negative ^3^
ZnO Z-Cote HP1	−3.09 ± 2.75	−6.76 ± 0.42	0.00	negative ^3^

^1^ Negative K- and C-peptide depletion values were set to 0.00 for the determination of the mean. ^2^ Mean peptide depletion values were evaluated according to the criteria mentioned in Table 2, regardless of chemical composition and solubility of the test substances and actual peptide—test substance ratios. ^3^ Considered “inconclusive” according to the test guideline as the test substance is inorganic and insoluble.

**Table 6 toxics-12-00616-t006:** Results and evaluation of the LuSens assay with test substances. Values of the calculated test substance concentration with 75% cell viability (CV75), maximum fold-induction values (Imax), and the calculated concentration with 1.50-fold luciferase induction (EC 1.50) are given, if applicable or determined.

Test Substance	CV75 [µg/mL]	Imax	EC 1.50 [µg/mL]	Result
CeO_2_ NM-212	532	0.71	n.a. ^1^	negative
BaSO_4_ NM-220	n.a. ^1^	1.41	n.a. ^1^	inconclusive ^2^
Levasil 200 (40%)	61.5	2.77	33	positive
SiO_2_ DQ12	498	3.32	<179 ^3^	positive
SiO_2_ Aerosil R972	n.a. ^1^	1.81	n.a. ^1^	inconclusive ^2^
SiO_2_ Aerosil 200	15	1.85	n.a. ^1^	negative
TiO_2_ NIST^®^ SRM^®^	351	1.01	n.a. ^1^	negative
TiO_2_ P25 Aeroxide	222	0.89	n.a. ^1^	negative
ZnO Z-Cote HP1	13.7	5.17	<7 ^3^	positive

^1^ n.a. = not applicable, no cytotoxic effects were observed in any of the concentrations, or the mean fold induction at the highest tested concentration (with a viability >70%) was lower than 1.50. ^2^ Considered “inconclusive” according to OECD TG442D as the test substance preparation was visually inhomogeneous (SiO_2_ Aerosil R972) or tested below the maximum concentration without cytotoxicity (BaSO_4_). ^3^ Not determined, the mean fold induction at the lowest tested concentration was higher than 1.50.

**Table 7 toxics-12-00616-t007:** Results and evaluation of the h-CLAT assay with test substances. Values of maximum fold-induction (Imax) and the calculated test substance concentrations resulting in a RFI of 150% for CD86 or 200% for CD54 (EC150 and EC200) are given, if applicable or determined.

Test Substance	EC150 (CD86)	Imax (CD86)	EC200 (CD54)	Imax (CD54)	Evaluation
CeO_2_ NM-212	n.a.	140	n.a.	100	negative/borderline for CD86 in one run ^1^
BaSO_4_ NM-220	n.a.	121	n.a.	97	negative
SiO_2_ Levasil 200 (40%)	n.a.	87	n.a.	201	inconclusive ^2^
SiO_2_ DQ12	n.a.	176	<144 µg/mL	970	positive
SiO_2_ Aerosil R972	n.a.	138	n.a.	120	inconclusive ^2^
SiO_2_ Aerosil 200	n.a.	171	n.a.	235	inconclusive ^3^
TiO_2_ NIST^®^ SRM^®^	n.a.	128	n.a.	132	negative/borderline for CD86 in one run ^1^
TiO_2_ P25 Aeroxide	<357 µg/mL	213	n.a.	152	positive
ZnO Z-Cote HP1	n.a.	163	<144 µg/mL	2585	positive

^1^ Results of valid experiments yielded negative (without borderline range) and negative or borderline if the borderline ranges provided in GL 497 are considered. Hence, at least one more independent experiment would be needed to derive a conclusive prediction in the h-CLAT. ^2^ Considered “inconclusive” according to the test guideline as the maximum concentration was below the maximum stated in the guideline and the actually tested maximum concentration was not cytotoxic. ^3^ Without considering the borderline range, the first run was positive for CD86, the second negative for both, and the third run was positive for CD54. Hence, at least one more independent experiment would be needed to derive a conclusive prediction in the h-CLAT. n.a. = not applicable.

**Table 8 toxics-12-00616-t008:** Evaluation results and ITSv2 score of NAMs, as well as evaluation of the 2o3 DA with DPRA, LuSens, and h-CLAT assays for assessing the skin sensitization potential of test substances. Information from the OECD Toolbox is not available ^1^.

Test Substance	DPRA ^1^	LuSens ^2^	h-CLAT	Evaluation 2o3 ^5^
CeO_2_ NM-212	n ^1^ inconclusive	n	n/br inconclusive	non sensitiser ^5^ inconclusive
BaSO_4_ NM-220	n ^1^ inconclusive	inconclusive ^2^	n	non sensitiser ^5^ inconclusive
SiO_2_ Levasil 200 (40%)	n ^1^ inconclusive	p	inconclusive ^3^	inconclusive
SiO_2_ DQ12	n ^1^ inconclusive	p	p	sensitiser
SiO_2_ Aerosil R972	n ^1^ inconclusive	inconclusive ^2^	inconclusive ^3^	inconclusive
SiO_2_ Aerosil 200	n ^1^ inconclusive	n	inconclusive ^4^	non sensitiser ^5^ inconclusive
TiO_2_ NIST^®^ SRM^®^	n ^1^ inconclusive	n	n/br inconclusive	non sensitiser ^5^ inconclusive
TiO_2_ P25 Aeroxide	n ^1^ inconclusive	n	p	non sensitiser ^5^ inconclusive
ZnO Z-Cote HP1	n ^1^ inconclusive	p	p	sensitiser

^1^ According to TG 442C, inorganic materials are out of the applicability domain of DPRA, and the insolubility of the NM makes the negative results inconclusive. ^2^ Considered “inconclusive” according to OECD TG 442D as the test substance preparation was visually inhomogeneous (SiO_2_ Aerosil R972) or tested below the maximum concentration without cytotoxicity (BaSO_4_). ^3^ Considered “inconclusive” according to the test guideline as the maximum concentration was below the maximum stated in the guideline and the actually tested maximum concentration was not cytotoxic. ^4^ Without considering the borderline range, the first run was positive for CD86, the second negative for both, and the third run was positive for CD54. Hence, at least one more independent experiment would be needed to derive a conclusive prediction in the h-CLAT. Considering the borderline range, the first run was borderline in both markers, and the following two runs were negative in CD86 and borderline in CD54. ^5^ Where two predictions are provided, the first evaluation disregards inconclusive results due to borderline evaluation in individual assays and limitations in the DPRA (see footnote 1 to this table). Abbreviations: n = negative; p = positive; br = borderline.

## Data Availability

Additional details are provided in the Appendix A.

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
