# Peer review of "Limitations and Modifications of Skin Sensitization NAMs for Testing Inorganic Nanomaterials"

_toxics, 2024, doi:10.3390/toxics12080616_

Round 1

Reviewer 1 Report

Comments and Suggestions for Authors

This is a work with practical research value. The work also provides a good reference and basis for future related research. Generally, this paper deserves to be published in the journal. The few comments facilitate the authors to further improve the manuscript.

In response to the conclusion drawn by the authors that the current NAMs which were developed for small organic molecules seem not to be a good fit for NM. On the one hand, the authors only validated 8 inorganic nanomaterials and did not validate other organic nanomaterials or carbon nanomaterials. Therefore, it is recommended that the authors provide the necessary clarification as to whether the unvalidated nanomaterials can produce biological effects similar to those of inorganic nanomaterials.

On the other hand, the authors need to give more valuable suggestions if the existing NAMs are not applicable. This provides constructive guidance for subsequent related studies.

Author Response

Dear Reviewer,

We would like to thank you for your thorough review.

Your comments (in black print) and our response (in blue print) are attached  

The manuscript and tables were changed accordingly.

Robert Landsiedel

Reviewer 2 Report

Comments and Suggestions for Authors

This manuscript evaluated the applicability and relevance of three skin sensitization NAMs described in OECD TG497. The study applied the NAMs to nine inorganic nanomaterials, prepared the NM samples following the NANOGENOTOX dispersion protocol, performed the DPRA, LuSens assay, and h-CLAT in accordance with OECD TG No.442C, D, and E, respectively, assessed skin sensitization using DAs as outlined in OECD Guidance 497. The study illustrated the limitations of these NAMs when applied to insoluble inorganic materials, highlighting issues such as lacking concentration metrics and unclear reaction mechanisms for skin sensitization caused by inorganic NMs. The manuscript concluded that these NAMs are not well-suited for the tested NMs.

This is a valuable study as it points out the limitations of applying current guidelines to inorganic NMs and identifies necessary improvements. However, the manuscript would benefit from deeper discussions. Therefore, I suggest considering to accept it after major revisions.

Below are my major concerns:

  • As the authors mentioned, the dispersion of the NMs is crucial, and stable dispersions should be applied to the test system. I suggest adding characterization of NMs in the medium, such as hydrodynamic size,and zeta potential.
  • One main reason for the "inconclusive" results is the limitations of the achievable tests. Stocks were prepared in this study following the NANOGENOTOX dispersion protocol. Please provide more explanation on the concentration selection for the three NAMs (20-fold/4-fold, 640 μg/mL, 1280 μg/mL, ratio to peptide).
  • Since all the NMs got "inconclusive" and inorganic materials are outside the applicability domain of DPRA, it is better to provide some literature discussion for possible alternative test methods. 
  • Line 403 mentions "No apparent influence of centrifugation was observed," while line 430 states "the h-CLAT assay was technically applicable with the addition of the centrifugation step." Please provide more explanation on the influence of centrifugation for the h-CLAT assay.

Below are my minor concerns:

  • The research focuses on NAMs for inorganic NMs, so please highlight this in the title.  
  • Since storage time can affect substance stability, please provide the storage time for stock substance.  
  • Please explain the calculation for the numbers mentioned in line 153. 
  • The color "highlight in orange" is not visible; please use another format to highlight.  
  • In Table 9, please double-check the h-CLAT evaluation against Table 8 for consistency, and add an introduction for the meanings of "0" and ">".  
  • It is better to provide a linked table of contents for the supplementary materials.
Comments on the Quality of English Language

English language is fine.

Author Response

(The authors gave the same response as above.)

Round 2

Reviewer 2 Report

Comments and Suggestions for Authors

I am satisfied with the author's responses to my questions and issues raised in my initial review. The author has answered my questions and addressed my concerns in the revised version. I recommend to accept with current version.